# The Relationship Between Academic Delay of Gratification and Depressive Symptoms Among College Students: Exploring the Roles of Academic Involution and Academic Resilience

**DOI:** 10.3390/bs15111486

**Published:** 2025-10-31

**Authors:** Xiaoli Ye, Wei Yang, Tingting Cheng, Haohao Gao

**Affiliations:** 1Institute of Education, Xiamen University, Xiamen 361005, China; 2Institute of Higher Education, Anhui University, Hefei 230039, China; 3College of Photonic and Electronic Information Engineering, Wuhan Technical University, Wuhan 430074, China

**Keywords:** academic delay satisfaction, depressive symptoms, academic involution, academic resilience, Chinese college students

## Abstract

In an era of rapid social transformation and hyper-competition, students in higher education are confronted with tremendous academic pressure, which is exacerbating mental health challenges at an alarming rate. This study used 576 Chinese college students as samples to construct and verify a moderated mediation model. The purpose was to systematically explore the associations among academic delay of gratification, depressive symptoms, academic involution, and academic resilience. The results demonstrate that academic delay of gratification exhibits a significant positive correlation with academic involution. Academic delay of gratification is negatively directly correlated with depressive symptoms, although it also exhibits a significant positive indirect correlation with depressive symptoms through academic involution. The mediating role of academic involution manifests as a suppression effect. Academic resilience is an important moderating variable. Low academic resilience intensifies the association between academic delay of gratification and academic involution. High academic resilience weakens this association. These findings not only elucidate the specific mechanism underlying academic delay of gratification and depressive symptoms but also provide a practical foundation for educational practitioners to develop effective intervention strategies.

## 1. Introduction

A pronounced global upsurge in psychological disorders has been observed over the past years, with college students exhibiting a high incidence rate ([66]). As a population in a critical transitional period of life, college students commonly face multiple challenges, including academic competition, employment pressure, and interpersonal difficulties, which increase their susceptibility to psychological disorders ([52]). International research finds that approximately 20% to 30% of college students report symptoms of depression, exceeding rates observed in the general public ([48]). Evidence from multiple countries has also confirmed that college students exhibit a notably elevated susceptibility to depressive symptoms. For instance, [37] ([37]) observed a depression prevalence rate of 34.5% within British college students. [62] ([62]) reported depressive symptoms in 18.4% of Spanish college students and anxiety symptoms in 23.6%. A survey of American college students involved in undergraduate research found that 23.4% suffered from severe depression ([30]). Similarly, Chinese college students also face a severe risk of depressive symptoms. According to data from the National Mental Health Development Report of China, individuals aged 18 to 24 exhibit markedly higher levels of depression than all other age demographics ([21]). The World Health Organization (WHO) forecasts that depression will become the primary factor of mental illness globally by 2030 ([72]). This trend is alarming, as depression not only causes cognitive impairment, slowed thinking, decreased concentration, and depressed mood ([1]) but also hinders college students’ participation in academic and social activities and, in extreme cases, may even result in suicide ([76]). Given these consequences, identifying actionable and protective elements that can mitigate depressive symptoms is crucial.

In China, with the popularization of higher education, a large number of graduates have flooded the labor market, intensifying employment competition ([58]) and increasing academic pressure on college students ([71]). With limited job opportunities, this competitive situation forces students to invest more energy in academic improvement in order to stand out among applicants. In this process, academic delay of gratification (ADOG) is an important predictor in relation to students’ academic achievements ([8]). Academic delay of gratification is the student’s psychological capacity to forgo short-term rewards in favor of achieving more meaningful and distant academic objectives ([7]). For college students in particular, appropriate academic delay of gratification can help balance the relationship between study and life, thereby promoting positive mental health development ([11]). However, there is a contradiction in the current higher education environment: when college students employ academic delay of gratification to overcome competition, they may be forced to continuously increase their academic input, leading to academic involution and subsequent continuous depletion of their psychological resources ([46]). Although existing research has affirmed the promoting function of delay of gratification on psychological well-being ([60]), evidence of how academic delay of gratification correlates with college students’ mental health in an intensely competitive academic setting remains limited. How academic involution functions between academic delay of gratification and depressive symptoms remains an area that requires further exploration.

Furthermore, the existing literature has established that academic resilience is a critical psychological element influencing college students’ learning experiences and results ([31]). As a psychological protective factor, resilience enables individuals to quickly recover from negative emotions ([2]). This indicates that academic resilience may mitigate the negative impacts of excessive competition. Previous studies offer limited insight into the relationship among academic resilience, academic involution, and academic delay of gratification. Evidence regarding the associations linking these factors to depressive symptoms is relatively scarce. Based on this gap, this study investigates how academic delay of gratification is associated with depressive symptoms and specifically evaluates the roles of academic involution and academic resilience. Our investigation aimed to reveal the association mechanism among academic delay of gratification, academic involution, academic resilience, and depressive symptoms through empirical analysis. The research results are intended to enrich academic discussions pertaining to the interplay between academic delay of gratification and depressive symptoms, making theoretical contributions to promoting college students’ psychological health and guiding them to establish a rational concept of academic competition.

## 2. Literature Review and Theoretical Hypothesis

### 2.1. Academic Delay of Gratification and Depressive Symptoms

[7] ([7]) were the first to introduce the notion of academic delay of gratification (ADOG), which is based on the application of the general framework of delay of gratification to the field of education. Academic delay of gratification denotes students’ capacity to put off chances for instant satisfaction as they prioritize the pursuit of key academic incentives or objectives, which are more temporally remote yet apparently of greater worth ([6]). [7] ([7]) created and used the Academic Delay of Gratification Scale (ADOGS), demonstrating a broad correlation between academic delay of gratification and two key characteristics of students: their motivation to achieve and their utilization of learning strategies. Research findings reveal that students who have a strong tendency for academic delay of gratification exhibit greater persistence when faced with less interesting or more challenging academic tasks ([4]). Moreover, compared to students who easily succumb to immediate impulses and temptations, those who can effectively employ academic delay of gratification are generally regarded as having higher intelligence, better academic performance, and social adaptability ([6]).

For college students striving for academic accomplishment, the significance of academic delay of gratification is clearly apparent. This research explores how academic delay of gratification relates to depressive symptoms among college students by reviewing previous empirical analyses and results. Within the scientific research field of delay of gratification, the findings of longitudinal studies conducted by Mischel’s team show that delay of gratification can not only help people cope with stress and emotional issues but also holds significant value for personal development ([55], [56]). Research shows that academic delay of gratification is a type of self-control requiring self-regulation, reflecting students’ firm willpower to withstand temptations ([5]), and that self-control ability is closely related to mental health status ([9]; [17]). Existing research findings have demonstrated that students with strong self-control ability have a lower probability of facing emotional problems, while individuals with weak self-control ability face an elevated risk of suffering from mental health disorders ([12]; [26]). The insufficiency of academic delay of gratification is alternatively termed present bias. People with present bias tend to be more impatient and willing to make immediate decisions ([61]). From this, we infer that college students who exhibit low tendency toward academic delay of gratification often have weak self-control and are more likely to experience psychological distress when facing difficulties and subsequently develop depressive symptoms ([24]; [60]). From the above discussion, we can draw the conclusion that academic delay of gratification and college students’ depressive symptoms are negatively correlated. Therefore, we propose Hypothesis 1:

**H1.** 
*Academic delay of gratification is negatively related to depressive symptoms among college students.*


### 2.2. The Suppression Role of Academic Involution

The concept of “involution” was initially proposed by the American anthropologist [27] ([27]) and refers to the situation where a cultural form, after reaching a stable stage, no longer graduates to new dimensions of development but instead turns to internal refinement. Subsequently, [25] ([25]) introduced this concept into agricultural research to describe the phenomenon where, due to population growth, labor is continuously invested but production efficiency does not increase. With the passage of time, the concept of involution has gradually extended from the social and economic fields to the educational field, referring to the excessive learning investment behavior of college students in pursuit of limited educational resources or opportunities ([45]). In this study, we define academic involution as the excessive resources beyond reasonable learning needs invested by college students in the competition for limited academic resources to gain advantage. Firstly, academic delay of gratification may exacerbate college students’ academic involution. Essentially, whether an individual tends to delay gratification or not is various behavioral manifestation of self-control ([23]). Research indicates that individuals with a tendency to delay of gratification often exhibit good adaptability in certain aspects, but they also tend to tightly control themselves and easily lapse into practicing unnecessary self-restraint ([22]). Although academic delay of gratification may help college students achieve excellent academic performance ([18]; [14]), excessive self-restraint and control can lead to excessive learning investment, thereby causing students to unconsciously promote or participate in academic involution ([50]). Existing research, utilizing latent growth models, has found that students with a stronger sense of academic value tend to invest unlimited time and energy in pursuit of excellent academic performance, thus falling into a cycle of involution ([80]). [77] ([77]) discovered that when students consider exam preparation to be of higher value than recreational activities, they will voluntarily reduce their recreational time and increase their learning investment. Related research has also pointed out that individuals with a strong tendency towards delay of gratification are more willing to invest a large amount of time and energy to achieve longer-term and more valuable goals ([63]). Therefore, college students who exhibit a strong tendency toward academic delay of gratification are more prone to over-invest time and energy in their studies ([6]), objectively promoting academic involution. In other words, an excessively high level of academic delay of gratification can exacerbate academic involution.

Secondly, as academic involution grows more noticeable among college students, their depressive symptoms tend to grow more severe. Academic involution is essentially “excessive competition”, distinguished from benign competition by its zero-sum game characteristics, which often leads to excessive academic pressure and resource depletion ([78]; [73]). Moderate academic competition may encourage college students to perform well academically and motivate them to develop themselves ([46]). However, academic involution differs from ordinary competition, often imposing excessive academic pressure on students and thereby damaging their psychological health and overall well-being ([50]; [45]). The cognitive appraisal theory of stress indicates that when individuals assess environmental events as potential threats to their well-being and perceive their coping resources as insufficient, it may trigger negative emotional, behavioral, and physiological responses ([42]). Academic involution can be regarded by college students as an external environmental event that threatens their development. This vicious competition can easily bring significant psychological pressure to individuals, leading to the accumulation of anxiety and helplessness and ultimately causing depressive symptoms. Existing empirical studies further support this mechanism. Research by [74] ([74]) uncovered that academic involution shares a significant positive correlation with both anxiety-related symptoms and depressive symptoms. [15] ([15]) found that involution intensifies the competitive atmosphere among students, and such intensification might act as a positive predictor for the increased stress experienced by students. Additionally, some studies suggest that academic involution among college students may lead to an “over-investment culture”, forcing them to shoulder an excessive academic burden, which in turn can cause anxiety, confusion, and depression ([47]). These studies collectively indicate that academic involution has emerged as a key factor posing a threat to college students’ mental health. Furthermore, although academic delay of gratification may directly alleviate depressive symptoms, it simultaneously increases the risk of depressive symptoms by intensifying academic involution. In this case, academic delay of gratification weakens its direct effect on depressive symptoms through the indirect influence of academic involution, potentially leading to a suppression effect. Drawing on the aforementioned analysis, this study put forward the following hypotheses:

**H2a.** 
*Academic delay of gratification is positively related to academic involution.*


**H2b.** 
*Academic involution is positively related to depressive symptoms.*


**H2c.** 
*The mediating role of academic involution between academic delay of gratification and depressive symptoms constitutes a suppression effect.*


### 2.3. The Moderating Role of Academic Resilience

Resilience, as a positive psychological resource, serves to mitigate the negative outcomes induced by other risk factors ([79]). Therefore, alongside the examination of academic involution’s mediating effect, the present research extends to include an exploration of academic resilience’s moderating effect. Academic resilience represents the good psychological state and adaptive ability that students demonstrate when facing academic challenges and setbacks ([53]), a concept derived from the study of psychological resilience. [64] ([64]) describe psychological resilience as an individual’s ability to adapt positively when confronting adversity, emphasizing its context specificity, that is, how it manifests differently in different environments. Academic resilience specifically denotes the manifestation of an individual’s psychological resilience in the educational context ([13]). According to resilience theory, resilient individuals possess the ability to prevent or overcome negative psychological outcomes when facing adversity and pressure ([43]). This ability can be regarded as the process of using internal resources to cope with external stressors and achieve self-adjustment ([29]). In other words, when facing academic pressure, students with academic resilience can reverse academic setbacks and failures and achieve success ([53]), while students who are lacking in academic resilience tend to face an imbalance between studies and personal life, thereby triggering mental health problems ([16]).

A substantial number of studies have established that resilience is able to moderate the influence of various psychological variables on individual behavior. [59] ([59]) found that resilience can weaken the impact of adverse emotions like anxiety and depression on internet addiction. Similarly, [75] ([75]) discovered that resilience functions as a pivotal moderating variable between self-control and academic procrastination, with resilient students being better able to withstand academic challenges and persist in completing academic tasks. These studies collectively reveal the key protective role of resilience in the individual’s psychological adaptation process. Based on these research findings, we draw the following theoretical inference. Students who demonstrate high academic resilience tend to possess better self-control abilities ([20]), and can pursue academic goals in a more flexible and adaptive manner, rather than becoming trapped in irrational competition and involution. Delay of gratification is widely regarded as a quintessential behavioral manifestation of self-control ([44]). In contrast, students with low academic resilience often exhibit insufficient self-control ([32]), making it difficult for them to balance the need for immediate gratification and long-term learning goals when facing academic pressure. Although these students may force themselves to delay gratification to increase their learning engagement, due to an inability to flexibly apply self-control strategies, they tend to fall into a rigid and excessive competition mode. This excessive suppression of psychological needs is actually a maladaptive academic delay of gratification ([65]). Therefore, this study posits that for less resilient students, the interplay between academic delay of gratification and academic involution is potentially strengthened. Consequently, the following hypothesis is derived:

**H3.** 
*Academic resilience moderates the association between academic delay of gratification and academic involution.*


### 2.4. The Present Study

Guided by existing theoretical and empirical evidence, our investigation constructed a hypothesized model, the structure of which is presented in Figure 1. This present exploration examines the association between academic delay of gratification and depressive symptoms, focusing on elucidating the underlying mechanisms. Specifically, we assess whether academic involution functions as a mediator linking academic delay of gratification to depressive symptoms and how this mediated pathway is moderated by varying levels of academic resilience. Unlike previous studies that predominantly emphasize the adaptive benefits of academic delay of gratification, our inquiry further explores its dynamic association with depressive symptoms within a highly competitive educational ecosystem. The findings are expected to make a valuable theoretical contribution to our understanding of the mechanisms linking academic delay of gratification to mental health outcomes while offering practical guidance for cultivating rational academic competition mindsets and promoting work–life balance among college students.

## 3. Methods

### 3.1. Data Source

Participants were recruited via convenience sampling from three universities in Anhui and Hubei Provinces, China, as research subjects. A questionnaire survey was administered, covering five sections: personal information, academic delay of gratification level, academic involution situation, academic resilience level, and assessment of depressive symptoms over the recent period. Data collection was conducted from March to June 2025 through the most widely used domestic questionnaire platform, Wenjuanxing, in an anonymous manner. All participants gained access to the survey using a unified electronic questionnaire link that was distributed to them, allowing them to choose a suitable time for completion. Our investigation strictly adhered to ethical norms, fully adopting a voluntary and anonymous form. All participants read the informed consent form and then confirmed their agreement by proceeding to fill out the questionnaire. They could withdraw at any time before or after participating. We solemnly promised all participants that their data would be utilized strictly for scientific research and their privacy will be protected without exception.

In total, 638 college students completed the questionnaire survey. Before conducting statistical analysis, a standardized data cleaning procedure was rigorously implemented to ensure data quality. The process consisted of two main steps. First, questionnaires with implausibly short response times, obvious input errors, or excessive missing values were excluded. Second, we further removed cases with non-random missing patterns or abnormally high rates of repeated responses, as these were identified as invalid or of low quality. After screening, 576 samples were deemed valid, for a recovery rate of 90.3%. All data processing procedures strictly adhered to the principle of confidentiality to safeguard participants’ privacy. Table 1 presents in detail the participants’ information. In the sample, 35.3% of participants were male, and 64.7% were female; 80.5% were undergraduates, and 19.5% were postgraduates; 54.9% were from “Double First-Class” universities (the “Double First-Class” designation is a national strategy launched by the Chinese government to develop world first-class universities and first-class academic discipline), and 45.1% were from other universities. The proportions of the surveyed students majoring in humanities and social sciences and natural sciences were 62.2% and 37.8%, respectively.

### 3.2. Measurement

#### 3.2.1. Academic Delay of Gratification

We employed the Chinese version of the academic delay of gratification scale revised by [41] ([41]). It was originally created by [7] ([7]) but was modified to suit the Chinese context, making it more suitable for the educational situation of Chinese college students. It consists of 10 items and comprises two dimensions. Factor one is “classroom academic delay of gratification”, reflecting students’ capacity to delay gratification during formal classroom learning. Factor two is “after-class academic delay of gratification”, pertaining to their after-class autonomous learning. All items present dilemma situations and are scored using a 4-point Likert scale (1 = definitely choose A; 2 = probably choose A; 3 = probably choose B; 4 = definitely choose B). Higher scores indicate a stronger tendency towards academic delay of gratification. Confirmatory factor analysis yielded the following model fit indices: X^2^/df = 4.033, RMSEA = 0.073, NFI = 0.938, TLI = 0.933, GFI = 0.953, IFI = 0.952, CFI = 0.952, and SRMR = 0.043. This scale’s Cronbach’s α coefficient was 0.866, suggesting that it has good reliability and validity.

#### 3.2.2. Academic Involution

The Academic Involution Scale for College Students in China (AISCSC), devised by [69] ([69]), was employed to evaluate the degree of academic involution exhibited by the Chinese college students. The scale comprises 16 items subdivided into three dimensions: social activities (5 items), assessing involution behaviors across lectures, student clubs, and internships; academic behavior (7 items), measuring involution-related behaviors within the learning environment; and social interaction (4 items), and evaluating involution behaviors associated with the establishment of interpersonal relationships, particularly with mentors, faculty advisors, and roommates. All items are measured using a 5-point Likert scale, ranging from 1 (strongly disagree) to 5 (strongly agree). A higher score is indicative of a more pronounced state of academic involution. Confirmatory factor analysis demonstrated acceptable fit indices (X^2^/df = 4.391, RMSEA = 0.077, NFI = 0.928, TLI = 0.931, GFI = 0.909, IFI = 0.943, CFI = 0.943, SRMR = 0.048) for the scale. Cronbach’s α coefficient was 0.935, indicating good reliability and validity of the scale.

#### 3.2.3. Academic Resilience

We used the six-item scale designed and validated by [53] ([53]) (“I think I am good at coping with academic pressure”) to assess students’ academic resilience. Each item was evaluated using a 5-point Likert scale, ranging from 1 (“strongly disagree”) to 5 (“strongly agree”). Elevated scores signify a heightened degree of academic resilience. The results of the confirmatory factor analysis manifested acceptable model fit, with specific indices as follows: X^2^/df = 4.300, RMSEA = 0.076, NFI = 0.977, TLI = 0.970, GFI = 0.978, IFI = 0.982, CFI = 0.982, SRMR = 0.024. The Cronbach’s α coefficient of this scale was 0.877, confirming that the scale had good reliability and validity.

#### 3.2.4. Depressive Symptoms

The depression subscale of the Depression Anxiety and Stress Scale (DASS-21), developed by [51] ([51]) and revised by [28] ([28]), was adopted to measure the depressive symptoms of Chinese college students. It contains 7 items that specifically focus on assessing depressive symptoms. It employs a 4-point Likert scale ranging from 1 (“not true”) to 4 (“always true”). In our research, we focused exclusively on the depression dimension. Increased scores point to a more serious level of depressive symptoms. Confirmatory factor analysis produced the following model fit indices: X^2^/df = 4.724, RMSEA = 0.080, NFI = 0.979, TLI = 0.971, GFI = 0.971, IFI = 0.983 CFI = 0.983, and SRMR = 0.029. Cronbach’s α coefficient was 0.913, confirming the scale’s good reliability and validity.

### 3.3. Data Analysis

For data analysis, demographic characteristics and descriptive statistics of the four primary variables in our investigation were obtained using SPSS 27.0, and a correlation matrix was generated. Subsequently, following [34]’ ([34]) recommendations, we tested the mediating effects and conducted moderated mediation model tests using models 4 and 7 in SPSS PROCESS.

## 4. Results

### 4.1. Common Method Bias Test

Given that the participants’ measurement data were collected using multiple scales, there was a potential for common method bias. Prior to formal data analysis, an assessment of potential common method variance was conducted via Harman’s single-factor test. We performed exploratory factor analysis in SPSS 27.0 by applying unrotated principal component analysis to all 39 measurement items in accordance with Harman’s single-factor test procedure. Seven common factors with eigenvalues greater than 1 were identified. The first factor accounted for 24.96% of the total variance, falling below the critical threshold of 40%. Thus, statistical tests revealed that common method bias was not a significant issue in this data.

### 4.2. Descriptive Statistics and Correlation Analysis

The means, standard deviations, and Pearson correlation coefficients of the four primary variables in this study are displayed in Table 2. Data from the correlation analysis suggest that academic delay of gratification exhibits a significant positive association with academic involution (r = 0.333, *p* < 0.01) and is significantly positively associated with academic resilience (r = 0.130, *p* < 0.01) and significantly negatively correlated with depressive symptoms (r = −0.162, *p* < 0.01). Academic involution is significantly positively related to academic resilience (r = 0.323, *p* < 0.01). Academic resilience displays a significant correlation with depressive symptoms (r = −0.130, *p* < 0.01). Analyzing the correlations yielded findings that buttress the preliminary hypotheses of this investigation.

### 4.3. Testing for Mediation Effect

To assess the mediating effect of academic involution, we first used model 4 in SPSS PROCESS created by [34] ([34]). During the analysis, the participants’ gender, educational level, type of universities, and major were incorporated as control variables to enhance the reliability of the outcomes. Further statistical details are provided in Table 3. The results show that academic delay gratification in model 1 was significantly negatively correlated with depressive symptoms (β = −0.167, t = −3.779, *p* < 0.001), providing support for H1. Following the inclusion of academic involution as a mediator, a significant positive correlation was found in model 2 between academic delay of gratification and academic involution (β = 0.398, t = 8.721, *p* < 0.001), supporting H2a. In model 3, academic involution showed a significant positive association with depressive symptoms (β = 0.106, t = 2.619, *p* < 0.01), verifying H2b. Meanwhile, academic delay in gratification showed a significant negative correlation with depressive symptoms (β = −0.209, t = −4.467, *p* < 0.001). The total, direct, and indirect effects of the mediation model are presented in Table 4. It is evident that a significant direct effect of academic delay of gratification on depressive symptoms was maintained, with a 95% CI ranging from −0.301 to −0.117, excluding 0. Statistical evidence supported the significance of academic involution’s indirect effect, producing a 95% CI that ranged from 0.003 to 0.082, which does not encompass 0. The total effect of academic delay of gratification on depressive symptoms was statistically significant, with a 95% CI of [−0.254, −0.080] that excludes 0. Notably, the direct effect and the indirect effect exhibit opposing signs, and the total effect demonstrates a reduction in magnitude compared to the direct effect, indicating that the total effect is suppressed. The suppression effect of academic involution was verified to be statistically significant, supporting H2c, with a suppression effect size of 20.1% (Figure 2).

### 4.4. Moderated Mediation Effect Analysis

Subsequently, to verify the moderating effect of academic resilience, a statistical assessment was performed using model 7 in SPSS PROCESS created by [34] ([34]). The results are presented in Table 5. By incorporating academic delay of gratification, academic resilience, and the interaction term (academic delay of gratification × academic resilience) into the model and controlling for gender, educational level, type of universities, and major, we investigated the moderating role of academic resilience. It was demonstrated that the interaction of academic delay of gratification and academic resilience was significantly correlated with academic involution (β = 0.106, *p* < 0.05), suggesting a moderating role of academic resilience in the association between academic delay of gratification and academic involution, thereby providing support for H3.

To further explore the interaction effects among the variables, we employed the simple slope analysis method to assess the strength of the moderating role of academic resilience. Additionally, to more clearly demonstrate the moderating effect of academic resilience, we plotted association curves depicting the link between academic delay of gratification and academic involution at different resilience levels, as depicted in Figure 3. The results manifested that when academic resilience was low (M − 1 SD), academic delay of gratification and academic involution exhibited a significant positive correlation (B_simple_ = 0.435, t = 7.138, *p* < 0.001). When academic resilience was high (M + 1 SD), although academic delay of gratification still had a statistically significant positive linkage with academic involution, the effect size decreased (B_simple_ = 0.280, t = 4.974, *p* < 0.001), indicating that the correlation between academic delay of gratification and academic involution weakened as academic resilience increased. Finally, the indirect effects of academic involution at different levels of academic resilience were examined, with the findings detailed in Table 6. At a low level of academic resilience, the indirect effect was 0.046, and the 95% CI was [0.003, 0.094]. At a high level of academic resilience, the estimated indirect effect was 0.030, bounded by a 95% CI ranging from 0.002 to 0.063, suggesting that as academic resilience levels increased, the mediating effect of academic involution remained significant, but its intensity decreased.

## 5. Discussion

This study developed a moderated mediation model to probe the direct and indirect links between academic delay of gratification and depressive symptoms in Chinese college students. Three key results emerged: First, academic delay of gratification shows a significant negative correlation with depressive symptoms. Second, academic involution mediates the association between academic delay of gratification and depressive symptoms, and there is a certain suppression effect. Third, academic resilience significantly moderates how academic delay of gratification relates to academic involution. These findings reveal the potential mediating and moderating mechanisms among academic delay of gratification, depressive symptoms, academic involution, and academic resilience, providing a theoretical basis for formulating effective interventions to enhance college students’ mental health.

### 5.1. The Relationship Between Academic Delay of Gratification and Depressive Symptoms

In this research, we not only explored the direct connection between academic delay of gratification and depressive symptoms but also incorporated academic involution into the analytical framework to further examine the relationship between the two. The findings indicate that among college students, both the total effect and direct effect of academic delay of gratification exhibit a significant negative correlation with depressive symptoms. However, after including academic involution, the indirect effect shows a positive correlation. This indicates that academic delay of gratification itself has the potential to alleviate depressive symptoms, but it may indirectly weaken its positive function by exacerbating academic involution. In other words, to a certain extent, academic delay of gratification can alleviate depressive symptoms. The outcome of this investigation is in line with prior studies ([4]; [17]). Previous research has shown that students with strong academic delay of gratification ability can plan their learning tasks more rationally and avoid the accumulation of stress caused by procrastination or cramming ([6]). Moreover, people with a strong ability to delay gratification often have better emotional control, and this self-control ability reduce academic anxiety and help maintain mental health ([9]; [35]; [24]). This study’s results also provide new evidence supporting the Conservation of Resources theory. This theory posits that individuals cope with stressful situations by obtaining and maintaining psychological resources ([36]), and academic delay of gratification, as a positive psychological resource, can help college students effectively manage their learning tasks and regulate their emotions, thereby alleviating academic pressure and reducing the occurrence of negative emotions. Most previous research has centered on the role of academic delay of gratification as a positive predictor of academic performance ([14]; [18]), neglecting the potential relationship between academic delay of gratification and the psychological health of college students. In light of the findings of the present study, educators and policymakers can integrate the cultivation of academic delay of gratification into the maintenance of mental health in multiple ways. Firstly, targeted self-management courses can be offered to systematically teach practical skills such as time planning and task decomposition, helping students enhance their academic delay of gratification ability while guiding them to arrange their study progress reasonably to avoid psychological stress caused by improper study planning. Secondly, institutions should help students recognize that excessive pursuit of delay of gratification may be associated with academic over-competition and assist them in finding a balance between persistence in achieving long-term goals and avoiding excessive self-exhaustion to prevent blind competition from damaging mental health. Thirdly, students can proactively develop self-control abilities to effectively improve their capacity to handle academic tasks and life challenges, thereby enhancing academic adaptability and laying a solid foundation for mental health.

### 5.2. The Suppression Effect of Academic Involution

The results of this study also indicate that the relationship between academic delay of gratification and depressive symptoms is mediated by academic involution, and this mediating effect manifests as a suppression effect. That is, the positive indirect influence of academic delay of gratification through academic involution, to some extent, weakens its negative total effect on depressive symptoms, thus partially suppressing the intensity of the total effect. This means that academic delay of gratification exerts its effect through more than a one-dimensional approach. It depends on the environment and the subsequent behaviors it triggers. In this study, it not only reduces depressive symptoms through its own regulatory ability but may also be “alienated” into a tool of “involution” in the context of an overly competitive environment, which could, in turn, transform it into a risk factor for depressive symptoms. Previous investigations established the presence of a notable positive correlation between academic delay of gratification and academic involution ([50]). Specifically, college students with greater academic delay of gratification ability tend to show excessive learning investment, exacerbating the phenomenon of academic involution ([80]). Clearly, academic involution can trigger depressive emotions in college students ([73]). Relevant research indicates that academic involution usually exerts negative effects on college students’ mental health, emotional state, and academic burden ([40]; [39]). Certain researchers further point out that academic involution, as an excessively competitive behavior, can cause individuals to bear excessive pressure and consume resources. Therefore, as the severity of college students’ academic involution increases, their depressive symptoms also tend to worsen ([70]). From this, it is evident that academic involution exerts an important mediating effect. To put it another way, students who possess a higher level of academic delay of gratification are theoretically more prone to succumb to academic involution, thereby triggering depressive symptoms. From a theoretical framework perspective, the findings of this research can be accounted for by the self-control energy model. This model argues that the energy required for self-control is a finite internal resource ([3]). College students who process a higher degree of academic delay of gratification need to continuously use self-control energy to fend off the expenditure of instant gratification and adhere to academic goals in the long run. However, academic involution intensifies the consumption of self-control energy in this process. When self-control energy is exhausted, college students may find it difficult to maintain high-intensity academic investment and competitive states, and they may feel mentally exhausted or exhibit a decline in vitality, thus experiencing negative emotions and depressive symptoms ([67]). This research indicates that while cultivating college students’ academic delay of gratification ability, we need to be vigilant about the possible problems of excessive competition and academic involution it may cause.

### 5.3. The Moderating Effect of Academic Resilience

As shown in Figure 2, our study further reveals that academic resilience is crucial for moderating the link between academic delay of gratification and academic involution. Specifically, compared with college students who possess high academic resilience, the connection between academic delay of gratification and academic involution is more significant among those with lower academic resilience. Our research findings suggest that academic resilience, as a key personal resource, can effectively buffer the relationship between academic delay of gratification and academic involution. This discovery corresponds with the Conservation of Resources Theory ([36]), an approach maintaining individuals’ fundamental motivation to obtain, maintain, and defend valuable resources. When individuals perceive a lack of resources or are unable to replenish them effectively, they enter a state of stress, which may lead to negative behaviors ([33]). In an academically competitive environment, students delay immediate gratification in favor of achieving long-term academic goals, thereby increasing their learning investment. However, when competition becomes involution, students with low academic resilience, due to their insufficient resource protection ability, are prone to continuously over-invest (such as staying up late to study) out of fear of falling behind, leading to cognitive resource depletion and emotional stress, and ultimately falling into academic involution. In contrast, students with high academic resilience can better maintain and replenish their psychological resources and avoid ineffective investment by adopting effective coping strategies (such as goal adjustment and stress management) ([10]). This finding verifies the core view of the resource conservation theory: individuals are capable of effectively managing stress when resources are sufficient, but resource imbalance leads to negative behaviors. Moreover, as a psychological variable closely related to personal mental health, academic resilience allows students to recover quickly and sustain an optimistic mindset to continue learning ([19]). Therefore, enhancing academic resilience can help mitigate the impact of academic delay of gratification on academic involution. Academic resilience, as a cultivable trait, should be given due attention in college mental health education ([68]). Especially in highly competitive environments, educators should have a deeper understanding of students’ needs and build multi-level social support systems, including peer assistance and psychological counseling. Through these measures, the negative impacts of competitive pressure can be effectively buffered, reducing the occurrence of academic involution from the source.

### 5.4. Research Implications

Our research employed a quantitative research approach to investigate the connection linking academic delay of gratification and depressive symptoms in college students, as well as the mediating effect of academic involution and the moderating function of academic resilience. The empirical analysis reveals the intricate interrelationships between academic delay of gratification, academic involution, academic resilience, and depressive symptoms, laying a theoretical foundation for future studies. Most prior studies hold that academic delay of gratification can have positive impacts on college students’ learning and development. This study, however, verifies the negative function of academic delay of gratification. Specifically, our investigation shows that academic involution acts as a suppressor, which means that academic delay of gratification indirectly heightens vulnerability to depressive symptoms by intensifying involution, partially offsetting its direct protective effect. That the suppression effect exists highlights how complex the connections are across academic delay of gratification, academic involution, and depressive symptoms, and it offers a more reasonable interpretation of the connection linking academic delay of gratification to depressive symptoms. That the suppression effect is present emphasizes the intricacy of the associations spanning academic delay of gratification, academic involution, and depressive symptoms, and offers a more rational explanation for the link connecting academic delay of gratification to depressive symptoms. The presence of the suppression effect highlights how complex the ties are across academic delay of gratification, academic involution, and depressive symptoms and provides a more reasonable account of the connection linking academic delay of gratification to depressive symptoms. Practically, our research furnishes a new perspective for promoting college students’ academic development and mental health. According to the research findings, academic delay of gratification has a “double-edged sword” effect on college students’ mental health. It is a protective factor against depressive symptoms while also potentially increasing the risk of depressive symptoms indirectly through intensifying involution.

Based on the research results, we offer some suggestions to college educators and policymakers. Firstly, colleges should cultivate students’ academic delay of gratification ability while establishing risk prevention mechanisms. Specifically, they can use “delay of gratification adaptive training” to help students master scientific goal decomposition and self-regulation strategies, effectively managing academic tasks and coping with academic challenges. Meanwhile, guiding students to carry out personal self-management happiness activities, such as recalling and elaborating on self-affirmation, goal setting or meaningful events every week, can effectively enhance the psychological well-being of college students ([57]). Additionally, colleges can develop learning efficacy assessment tools to quantitatively analyze the ratio of delay of gratification behavior to actual academic gains, promptly identifying ineffective efforts. Secondly, colleges need to introduce diverse assessment standards, incorporating academic performance, practical innovation, and mental health into the comprehensive assessment system to alleviate the pressure to intensively pursue academic achievements. Thirdly, educators should set clear goals and standards for students, providing them more opportunities to engage in learning and tackle challenges, thereby enhancing their academic resilience ([68]). Finally, in response to the increasingly fierce competition in the current education field, policy measures should be taken at both levels. The government should focus on promoting the balanced allocation of educational resources to fundamentally alleviate academic anxiety caused by structural educational inequality. At the same time, in the social and cultural contexts, it is necessary to disseminate diverse success concepts through multiple channels to gradually change the single “score-oriented” evaluation standard. These measures can not only preserve the positive value of academic delay of gratification in cultivating self-discipline and planning abilities but also effectively prevent its transformation into involution, thereby ensuring that college students maintain their mental health while pursuing academic success.

## 6. Limitations and Future Research

Several limitations of this research must be acknowledged, indicating fruitful avenues for future research. This investigation primarily utilized a cross-sectional design, and the collected data only reflected the participants’ state at the moment of the survey, meaning it was unable to draw accurate inferences about the causal relationship between academic delay of gratification and depressive symptoms. Additionally, although this investigation confirmed a positive correlation between academic delay of gratification and academic involution, this discovery needs to be understood in combination with the specific context of Chinese society. In China’s social and cultural setting, excellent academic performance is regarded as an effective method of achieving higher social status ([38]). Compared with the Western educational environment, Chinese students often place college entrance examinations at the core of their personal development and thus experience more significant academic pressure ([49]). This unique social and cultural background may lead to a special reinforcing mechanism in which academic delay of gratification and academic involution strengthen each other. Therefore, the universality and generalizability of the research results are somewhat limited. The unique value of our investigation lies in revealing the contradictory nature reflected by the “suppression effect”: in the Chinese cultural context, academic delay of gratification could potentially function as a “double-edged sword”. When college students have a moderate tendency to delay academic rewards, it can help them develop by reducing their depressive symptoms. Conversely, an excessive degree of academic delay of gratification might incur negative effects by intensifying academic involution. However, this study did not deeply explore just how effective academic delay of gratification is. In other words, it did not precisely delineate the “moderate scope” of academic delay of gratification. Finally, this study mainly focused on academic factors and failed to incorporate other significant sociocultural factors that might explain depressive symptoms among Chinese college students. For instance, the context of being an only child could lead to specific interpersonal alienation and loneliness for students in the collective university environment, which is also an important trigger for depressive symptoms. However, this research has not yet explored the interaction between these factors and academic mechanisms.

Future research can be improved in the following ways. Firstly, future research should adopt more refined experimental research or longitudinal designs. These methods can offer more explicit evidence regarding the causal relationships and mediating mechanisms between variables, so as to provide a basis for further verifying the various findings of this study. Secondly, it is necessary to conduct cross-cultural comparisons in different competitive models and social and cultural backgrounds to verify the universality and particularity of this relationship. Thirdly, subsequent investigations can set different levels of academic delay of gratification, combined with longitudinal tracking data, to analyze how academic delay of gratification specifically affects depressive emotions and academic involution in different scopes and determine the critical value at which it exerts a positive effect, providing more precise guidance for college students to reasonably enhance their academic delay of gratification ability in educational practice. Lastly, we suggested that future research integrate the family and cultural background of Chinese society to explore the diverse sociocultural factors influencing depressive symptoms among college students, with the aim of constructing a more comprehensive theoretical model and provide more targeted references for improving the mental health status of college students.

## 7. Conclusions

The current investigation examined the mechanism of academic involution in the connection between academic delay of gratification and depressive symptoms among Chinese college students and explored how academic resilience moderates the association between academic delay of gratification and academic involution, revealing the complex relationship among these variables. Our research established that academic delay of gratification was significantly negatively related to depressive symptoms. However, this protective effect was partially offset by the suppression effect of academic involution. In other words, academic delay of gratification indirectly increased the risk of depressive symptoms by intensifying academic involution. Notably, academic resilience played a key moderating role in this mechanism.

In general, this study not only reveals the “double-edged sword” effect of academic delay of gratification on mental health and its underlying mechanisms but also echoes the ongoing academic discussion about the complex relationship between academic engagement and academic performance ([54]). Based on this, we suggest that educators move beyond the single advocacy of academic delay of gratification ability and instead adopt comprehensive measures: while cultivating students’ goal management abilities, they should focus on creating a supportive teaching environment to alleviate institutional pressure and systematically enhance students’ academic resilience levels. This provides both theoretical and practical guidance for promoting the academic achievements and mental health of college students in a high-pressure academic context in a coordinated manner.

## Figures and Tables

**Figure 1 behavsci-15-01486-f001:**
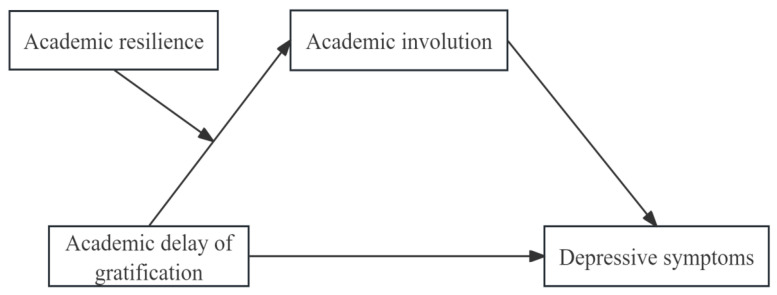
Hypothesized model.

**Figure 2 behavsci-15-01486-f002:**
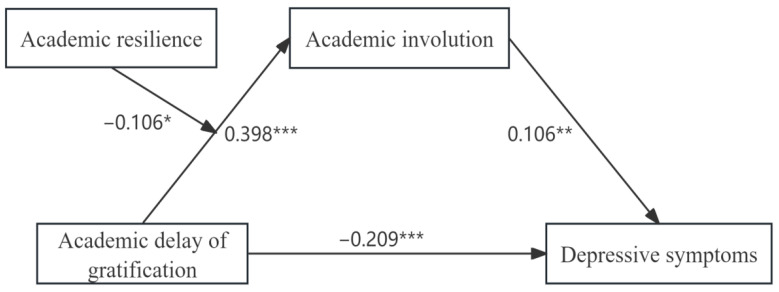
The moderated mediation model (note: * *p* < 0.05; ** *p* < 0.01; *** *p* < 0.001).

**Figure 3 behavsci-15-01486-f003:**
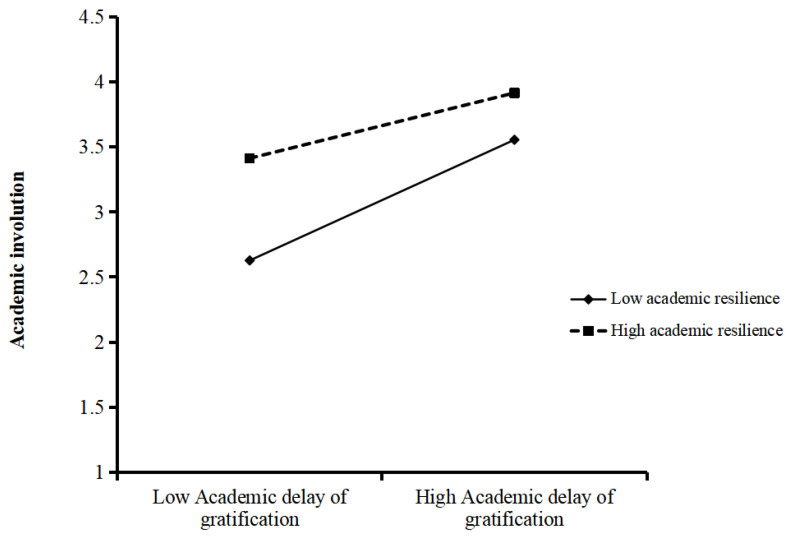
Moderating effect of academic resilience.

**Table 1 behavsci-15-01486-t001:** Sample demographics (n = 576).

Variable	Frequency (Percentage)
**Gender**	
Male	203 (35.3%)
Female	373 (64.7%)
**Education level**	
Undergraduate students	465 (80.5%)
Postgraduate students	111 (19.5%)
**Types of universities**	
Double First-Class universities	316 (54.9%)
Other universities	260 (45.1%)
**Majors**	
Humanities and social sciences	358 (62.2%)
Natural sciences	218 (37.8%)

**Table 2 behavsci-15-01486-t002:** Descriptive statistics and correlation analysis (n = 576).

Variables	*M*	*SD*	1	2	3	4
Academic delay of gratification	2.968	0.668	1			
Depressive symptoms	1.847	0.707	−0.162 **	1		
Academic involution	3.277	0.766	0.333 **	0.054	1	
Academic resilience	3.486	0.732	0.130 **	−0.130 **	0.323 **	1

Note: ** *p* < 0.01. M: mean. SD: standard deviation.

**Table 3 behavsci-15-01486-t003:** Testing the mediation effect.

	Depressive Symptoms (Model 1)		Academic Involution (Model 2)		Depressive Symptoms (Model 3)	
	b	t	b	t	b	t
Academic delay of gratification Academic involution	−0.167	−3.779 ***	0.398	8.721 ***	−0.209	−4.467 ***
					0.106	2.619 **
R^2^	0.036		0.123		0.048	
F	4.301 ***		16.052 ***		4.764 ***	

Note: ** *p* < 0.01, *** *p* < 0.001.

**Table 4 behavsci-15-01486-t004:** Decomposition of mediation effect.

	Effect	Boot SE	Bootstrap 95% CI
Direct effect	−0.209	0.047	[−0.301, −0.117]
Indirect effect	0.042	0.020	[0.003, 0.082]
Total effect	−0.167	0.044	[−0.254, −0.080]

**Table 5 behavsci-15-01486-t005:** Testing the moderated mediation effect.

	Academic Involution		Depressive Symptoms	
	b	t	b	t
Academic delay of gratification	0.357	8.082 ***	−0.209	−4.667 ***
Academic resilience	0.286	7.164 ***		
Academic delay of gratification × academic resilience	−0.106	−2.018 *		
Academic involution			0.106	2.619 **
R^2^	0.205		0.048	
F	20.854 ***		4.764 ***	

Note: * *p* < 0.05; ** *p* < 0.01; *** *p* < 0.001.

**Table 6 behavsci-15-01486-t006:** Decomposition of moderated mediation effect.

	Academic Resilience	Effect	Boot SE	Bootstrap 95% CI
Moderated mediation effect	eff1 (M − 1 SD)	0.046	0.023	[0.003, 0.094]
	eff2 (M)	0.038	0.018	[0.002, 0.074]
	eff3 (M + 1 SD)	0.030	0.016	[0.002, 0.063]

## Data Availability

The data that support the findings of this study are available from the corresponding author upon reasonable request.

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
