# Peer review of "The Relationship Between Academic Delay of Gratification and Depressive Symptoms Among College Students: Exploring the Roles of Academic Involution and Academic Resilience"

_behavsci, 2025, doi:10.3390/bs15111486_

Round 1

Reviewer 1 Report

Comments and Suggestions for Authors

The manuscript reports on the study examining the relationship between academic delay of gratification and depressive symptoms among college students in China. The authors specifically assess the mediating role of academic involution for this relationship and the moderating influence of academic resilience on the link between academic delay of gratification and academic involution.

This study offers valuable insight into the theoretical relationship between academic delay of gratification and depressive symptoms, thereby laying a crucial groundwork for practitioners to design strategies/supports targeting mental health issues (such as stress and depression) resulting from academic involution. The manuscript is well-organized and clearly written. I have only a couple of minor comments for the authors to consider to further enhance the manuscript prior to final publication.

p11, lines 419-420: Since the measure of academic resilience used a 5-point Likert scale, it would be beneficial to inform readers how resilience levels were categorized into the two groups (i.e., the cut-off score used to define high vs. low resilience). Also, it is recommended to include this information in the Methods—Measurement section.

p12, Figure 3: The Y-axis title in Figure 3 appears to be incorrectly labeled as 'Academic resilience' when it should likely be 'Academic Involution.' Please verify and correct this label.

Reviewer 2 Report

Comments and Suggestions for Authors

Greetings 

This was well written- however I have the following suggestions/raise the following concerns. 

  1. The intro can be shortened as the definition of academic delay of gratification is below.  Perhaps define it briefly 1-2 sentences so the intro and the beginning of the lit review are not so redundant.
  2. The defintion of academic involution is not clear- this section is not as tight as the defintion of academic delay of gratification and the paragraphs do not flow well here.  It would be good to have a crisp defintion of this. 
  3. Curious to know if the sample is representative of the population- could use some national data to check variables such as gender etc. 
  4. Modifying the scales to suit the Chinese context is important - but without testing the factor structure to assess if its consistent is ill-fated and short sighted. The two sentences on the CFA and alpha were not enough and the fit indices were not adequate.  I feel like that was swept under the rug.  Also why CFA and not EFA to test the validity of this chinese version?  Felt like explanations were missing. 
  5. Curious why used CFA.  The model showed 4 different latent constructs which would be a path analysis not a CFA which tests measurement models.  It seems like the analysis doesnt fit the model. 
  6. The CFAs were conducted and are presented very briefly describing each indiviudal measure- these are not presented fully and it feels off since the results are presented, but then the following paragraph is where you explain that CFA was conducted.  Then in the results section no CFA was done. 
  7. It would be best to reconfigure the results section to include the CFA of the measures, and EXPLAIN the results.  I think this is key especially since some fit indicies are not ideal and should be explained. 
  8. Perhaps presenting the results by hypothesis and RQs. 
  9. Depressive symptoms in Chinese students are not explained.  Why is this occurring.  There is comparison to American students, but this should be unpacked a little more in the Chinese context,  is it related to academic pressure, loneliness as most students are only children with lack of extended family support?  
  10. Conclusion could be expanded upon.  Feels short and abrupt and devoid of connections to the literature. 
Comments on the Quality of English Language

There are some turn of phrases that should be checked, some paragraphs that do not begin with topic sentence followed by supporting sentences.  

Overall it reads well, but there are small issues throughout. 

Reviewer 3 Report

Comments and Suggestions for Authors

The MS presents a research examining the relationships among academic involution, resilience and gratification delay in a sample of college students by also considerring depressive symptoms.

It is overall well written and structured. The method and results clearly presented and discussed. The factor involution is understudied and it is nice it has been considered.

Thus, I am positive towards publication pending a few revisions I am listing below

  1. Introduction.  I would have pointed out the choice of the variables and the exclusion criteria. For instance, they administered the DASS which includes a measure of stress and anxiety. Why not considered?
  2. The sample is rather equally represented by males and females.  The literature reports gender differences in some of the variables considered, e.g. depression levels. Thus, I would advice including gender in the analyses
  3. I am not sure that all the instruments were validated in the language they were delivered. Please specify and if not maybe this aspect should be reported in the limitations
  4. I would be more specific on the potential ways to sustain university students well-being by referring to existent programs, e.g. 'Fostering resilience among university students: the role of teaching and learning environments' and Does the Weekly Practice of Recalling and Elaborating Episodes Raise Well-Being in University Students?
  5. Overall,  I would avoid the causal language, e.g. 'predict' due to the cross-sectional nature of the study
  6. Something seems wrong in the Figure: the same variable is reported twice, as mediator and outcome (y axis)
  7. Please refer always to depressive symptoms not depression 

Round 2

Reviewer 2 Report

Comments and Suggestions for Authors

I appreciate the authors attention to detail and responses to the comments- I believe many of my earlier comments were based on an assumption of the purpose of the paper based on the language used which was misleading and fixing that language fixed a lot of the confusion on the methods.  I do believe the authors also addressed all my previous concerns and that the revisions make the paper better.